# Investigation on the Mechanism and Parametric Description of Non-Synchronous Blade Vibration

**DOI:** 10.3390/e23040383

**Published:** 2021-03-24

**Authors:** Mingming Zhang, Anping Hou, Yadong Han

**Affiliations:** 1Faculty of Science, Beijing University of Technology, Beijing 100124, China; mmzhang@bjut.edu.cn; 2School of Energy and Power, Beihang University, Beijing 100191, China; hanyadong@buaa.edu.cn

**Keywords:** axial compressor, non-synchronous vibration, fluid-structure interaction, limit cycle oscillation, inter phase angle, spiral vortex

## Abstract

In order to explore the mechanism during the process of the non-synchronous vibration (NSV), the flow field formation development is investigated in this paper. Based on the fluid–structure interaction method, the vibration of rotor blades is found to be in the first bending mode with a non-integral order (4.6) of the rotation speed. Referring to the constant inter blade phase angle (IBPA), the appearances of frequency-locking and phase-locking can be identified for the NSV. A periodical instability flow emerges in the tip region with the mixture of separation vortex and tip leakage flow. Due to the nonlinearities of fluid and structure, the blade vibration exhibits a limit cycle oscillation (LCO) response. The separation vortex presenting a spiral structure propagates in the annulus, indicating a pattern as modal oscillation. A flow induced vibration is initiated by the spiral vortex in the tip. The large pressure oscillation caused by the movement of the spiral vortex is regarded as a main factor for the presented NSV. As the oscillation of blade loading occurs with blade rotating pass the disturbances, the intensity of the reverse leakage flow in adjacent channels also plays a crucial role in the blade vibration.

## 1. Introduction

Aeroelastic problems in modern turbomachinery have been recognized as major factors for the failures of blades. Due to the interaction between the unsteady flow and the blade structure, the flow induced vibrations are widely in existence. In recent years an atypical phenomenon of blade vibration which is described as non-synchronous vibration (NSV) has attracted a lot of attention. It indicates that the frequency of blade vibration is not an integer order of the rotor frequency. It was early observed by Baumgartner et al. [1] that the large vibration of blade was found in the first bending mode on a multistage compressor. He pointed out the mechanism of instability in unsteady flow was the tip shedding leakage flow, as the main source behind NSV.

Then this phenomenon was also observed in other studies with the occurrence of NSV in a non-engine order of blade passing frequency [2,3,4,5]. The tip size is considered as a main source which influences this rotation instability by Mailach et al. [2]. The experiments conducted by Marz et al. [3] indicate that this vibration is caused by the fluctuating blade tip vortices when moving from the suction side to the pressure side in blade passages. It is shown that the NSV is caused by the vortex oscillation and the tip flow instability [4]. In the discussion of Sanders [5], the aerodynamic instability is induced by the interactions with the passage shock and the separation flow. The large amplitude periodic oscillation leads to non-synchronous vibration problems.

Related to rotation instability, a study of the formation of stall disturbances was carried out by Vo [6]. A link is made between the flow fluctuations and the formation of spike stall disturbances. A simple model was developed by Thomassin et al. to predict the rotor critical speed when NSV occurs based on the jet core feedback theory [7,8]. The explanation to NSV phenomenon is proposed as the blade tip trailing edge flow acting likes an impinging jet on the pressure side. Based on the principle of energy transport, blade vibration with an input motion is applied by Spiker et al. [9]. Blade mode damping is taken into consideration [10]. The limit cycle oscillation (LCO) is determined with a negative damping. The non-synchronous vibration is interpreted as a resonance with tip leakage flow by Drelet et al. [11,12] and a correlation as a function of the tip clearance size is shown to influence the critical NSV blade tip speed predictions significantly.

With the decreasing flow rate, the blockage of leakage flow happens at the tip. The blade vibration is attributed to the oscillating blade tip vortex by Hwang et al. [13]. By using the model of Kielb [14], studies have been carried out by Zha et al. [15,16,17,18,19,20] to explore the flow at tips. The results reveal that the tip clearance shape has an influence on the frequency of NSV [15]. The separation vortex of the tornado structure in the flow field and the tip leakage flow are the main factors of the blade non-synchronous vibration [17]. This kind of vortex travels on the suction side in the passage, crossing to the leading edge of the next blade in the counter rotating direction. Such a motion creates a pair of coupling forces due to two low pressure regions generated by the vortex positions. The oscillation by the vortex coming and leaving generates a torsion moment, coupling with the blade structure in modal vibration [18]. Simulations [19,20] coupled with fluid and structure studies indicate that the blade vibration exhibits a limit cycle oscillation excited by the aerodynamic force, but the comparison results also show that the first torsion mode observed in the rig test is accompanied by the first bending vibration of the blade, which is not obvious in the experiments.

It is concluded that the aerodynamic flow instability caused by the leakage vortex in the tip region is mostly considered to be the main factor responsible for non-synchronous vibrations [21]. The blockage at the tip leads to tip flow fluctuations near the stall region, which travel through the passage and impinge on the adjacent blade. In the analysis of tip flow blockage [22], the oscillation of separated flow in the tip is accounted for by the aeroelastic stability. By capturing the resonance of flow instability and blade vibration, a flow-induced vibration is initiated by the leakage flow at the tip [23] and hence the devolvement of the leakage flow in the process of vibration is needed for further study as pointed out by the authors.

Because of the importance of leakage flow on NSV, tip-timing analysis has been presented for measurements [24,25,26,27]. The mechanism of vibrations is based on an unsteady flow pattern that rotates relative to the rotor, as a result of multiple surge events [24]. This technique provides information about the blade vibration behaviors [25,26]. The results show the process of stall cell evolution and how blades vibrate when suffering such an aerodynamic load [27]. Besides becoming a hotspot of fluid mechanics, the acoustic mechanism has also been of interest. By establishing an analytical sound model [28], sound generation is linked with blade oscillation. The root cause for the oscillations is identified as an acoustic mode [29], which affects the amplitude of blade vibrations. A semi-analytical model based on single degree of freedom oscillators was developed by Stapelfeldt [30] to assess the sensitivity of NSV to driving parameters.

Despite the extensive research on this topic, the physical mechanism and the lock-in phenomenon in NSV are still not well understood. To describe a NSV encountered in a high speed compressor, a time domain numerical approach is carried out by the fluid-structure iterative coupling method. The aim of the current paper is to explore the origin and development of the entropy flow which causes blade vibrations and discuss the parametric description for NSV. This paper is organized as follows: Since the large amplitude oscillating leakage flow in the tip is regarded as a main inducing factor, the initial and developing process of reverse flow is described firstly in in Section 3 and the nonlinear responses of blades are analyzed. Secondly, the fluid mechanics for blade vibrations with the resonance of tip flow and blade are discussed in Section 4. Lastly, the parameterization of the inducing factors is preliminarily studied based on the flow structure evolution.

## 2. Descriptions of Numerical Methodology

In the experiments, a multistage high-speed axial compressor exhibits a large amplitude blade vibration within a certain speed range (9700 rpm~10,190 rpm). It is observed that there is a 2° deviation in the normal adjustment of the adjustable guide vane. This condition is shown to cause the compressor to deviate from the design working state. The blade vibration is found as in a frequency-locking state, which is close to the blade first bending mode with a non-integral order (4.4–4.6) of the rotation speed. This kind of blade vibration is considered to be a non-engine order of the rotor. The first 1.5 stages of the multistage transonic compressor with 42 inlet guide vanes, 38 rotor blades and 82 stator blades is adopted in this paper for the vibration investigation. A detailed introduction of the rig is presented by Zhang [23].

As the non-synchronous blade vibration phenomenon involves the coupling of unsteady flows and blade structures, the time-domain fluid structure coupling simulations [31] are solved in this paper. The full annulus configuration model is adopted to capture the detailed NSV with flow excitation. The commercial ANSYS CFD software package is used for the current calculation. The unsteady flow field is solved by using the numerical solution of 3-D Navier-Stokes equations by the commercial software CFX. The spatial discretization of the flow governing equations is employed on a high-resolution scheme, and second-order backward differencing is integrated for the time-accurate solution [32]. The total pressure boundary condition is used at the inlet and the averaged static pressure boundary condition is set to the outlet with a constant rotor speed. Smooth, adiabatic, and no-slip wall boundary conditions are applied for the flow field solution. The numerical model of the 1.5-stage transonic turbocompressor is shown in Figure 1.

In consideration of computation costs, the grid independence verification is made to achieve a balance between the computation efficiency and accuracy. The mesh refinement is made in the regions of the boundaries and tip because of the importance of the flow separations and tip leakage vortices. The turbulence model is used as the standard *k-ε* function for the wall treatment by dealing with a coarse grid [23], and the scalable wall function is applied to the modeling of the boundary layer. The verification for the grid independence from the mesh scale is conducted by a comparison through steady computations with different grid sizes. Y-plus of the meshes near the wall boundary is kept smaller than 20 in all cases. As reported from Figure 2, the quantitative mesh elements are approximately selected as 8.09 million based on performances with different grid densities and the mesh size is still kept relatively coarse to limit the computational cost. The distributions of the used cells in domains are provided in the Table 1. The blade natural mode frequency of the first bending is calculated to be 789 Hz at the rotating speed of 10,186 rpm. During the unsteady computations of fluid structure coupling, every rotor blade pitch is divided into 10 time steps, with each time-step including a maximum of 30 inner interaction steps. Then the time step is calculated to be 1.549 × 10^−5^ s. Convergent numerical results are obtained after at least five revolutions.

## 3. Investigation on the Flow Field Structure in Non-Synchronous Vibration

As the reliability and accuracy of the numerical method were verified earlier in [23], the comparison with the experiment demonstrates that it is sufficient to predict the NSV phenomenon by the current methodology. Thereafter, the focus on flow structure and mechanism in the non-synchronous vibration is mainly carried on by paying attention to the resonance of flow instability and blade vibration in consideration of the fluid–structure interaction.

### 3.1. Anlysis on Blade Vibration Characteristics under Different Working Conditions

In order to observe the vibration phenomenon in different states, a series of computations are conducted by changing the back static pressure implemented at the exit. The characteristic line of the compressor is shown in Figure 3 under the operating speed of 10,186 rpm.

For the sake of exploring the origin and development of NSV, the analysis on the vibration characteristics of compressor blade is performed under different working conditions. During the working conditions, NSV occurs in state C, and the displacement deformations of the blades in vibration at state C are drawn up in Figure 4. The deformation results indicate that the vibration of the blades exhibit the first bending mode with the maximum displacement deformation point arisen at the leading edge of the blade tip. Moreover, the blade obviously presents a different deflection with each other at the same time during the process of vibration. It means there may be a spatial harmonic composed in the circumferential direction. This appearance will be discussed in detail at another section.

The displacement during the vibration is observed according to the numerical monitor attached at the node of maximum deformation on the blade tip leading edge. To inspect the NSV with frequency variation, the spectrum analysis on the blade vibration is sketched in Figure 5. Due to the interaction with inlet guide vanes, the rotating frequency and its multiplied frequencies are captured clearly in the rotor domain. Besides these frequencies in multiples of the engine order (EO), the frequency *f_BV_* (blade vibration) of 789 Hz is shown in the blade vibration, which is around 4.6 times of the engine order. In the experiment, the vibration of the rotor blade in first stage was found to be in the first bending mode with a non-integral order (4.4–4.6) of the rotating speed. This is consistent with the phenomenon of the experiment. But this characteristic frequency doesn’t appear in a suddenness, but with a developing process. According to the description of reference [23], there is not a non-synchronous vibration when the compressor operating away from the state C (NSV point). As approaching to the stall boundary, the frequency of 4.6 EO starts to emerge gradually which is revealed in Figure 5a. With the further close to the boundary, the amplitude of this characteristic frequency increases rapidly to the dominant status (Figure 5b). The amplitude reaches the peak at state C shown in Figure 5c.

However, the blade vibration shows a reduction after that and the amplitude of the frequency in non-integral order finally decreases to a very low level, as sketched in Figure 5d. The development of NSV is represented in Figure 6 by plotting the variation of frequency amplitude in the non-integral order of 4.6 EO. The occurrence of a non-synchronous vibration is accompanied by the origin of this characteristic frequency and the magnitude of this kind of vibration increases continually. When the amplitude of the characteristic frequency reduces persistently, the blade vibration withdraws from the state of NSV.

### 3.2. Variation of Interblade Phase Angle during Vibration

In the process of blade vibration, there exists a phase angle between adjacent blades within a certain angle range, which is called interblade phase angle (IBPA). Based on the periodic symmetry assumption, the traveling wave theory was proposed in flutter. It is assumed that the vibration amplitude and frequency of each rotor blade are all the same, but with a fixed phase difference between the adjacent blades, but in the actual working state, the interphase angle of each blade is not exactly the same. In this part, the changes of IBPA in blade vibration are explored under the condition of entry and departure from a non-synchronous state.

The blade phase angles calculated during the vibration have been visualized circularly like a wave superimposed on the rotor in Figure 7. The blades deflects a certain angle in sequence from each other. As discussed in the previous section, the characteristic frequency of non-integral order appears since state A, and starts to be in a dominant position with state B. The average value of the inter blade phase angle is computed to be 151.5° at state B in Figure 7a for instance. According to the equation below:
(1)$$D=\frac{\alpha_{ibpa}}{2\pi }\times N,$$
where *D* represents the nodal diameter number, $\alpha_{ibpa}$ expresses the inter blade phase angle, *N* means the number of blades. The nodal diameter in this state is calculated as 16, which indicates that there is a traveling wave with a pitch diameter number of 16. Then the vibrating blades are affected by this wave propagating along the circumferential direction with different phase indexing. This appearance is still visible in Figure 4, which indicates the deformations of the vibration for the adjacent blades.

However, when the blades withdraw from the NSV, there is also an obvious change in the IBPA at state D. The phase angles present an evident discrete distribution displayed in Figure 7b, which is different from the former ones. The current state result showing up means that there is not a clear spatial harmonic composed by the superposition of a number of rotating nodal diameter patterns. The variation of the IBPA during blade vibration is represented in Figure 8. It can be seen that the average inter blade phase angle is maintained a close value as the rotor enters into the non-synchronous vibration state and the average IBPA decreases rapidly when the blades gradually bow out of the state in non-synchronous vibration. Referring to the constant IBPA corresponding to the modal vibration, the appearances of frequency-locking and phase-locking can be identified for the NSV with a non-integral order.

### 3.3. Nonlinear Dynamic Response of Blade Vibration

Due to the nonlinearities of the fluid and structure, the essence of blade vibration is a nonlinear dynamic system. Studies from the perspective of nonlinear dynamics show that the blade system is usually accompanied by the generation of limit cycle oscillations (LCOs). In this part an effort has been devoted to the LCO response analysis of the non-synchronous vibration with the phase planes of blade tip using dynamic system stability theory.

The response of the system can be divided into three stages according to the phase diagrams plotted in Figure 9. In the first stage, the blade oscillates due to the excitation of shedding vortex resulting the flow-induced vibration of blade. The phase image of the blade vibration presents to be in chaotic state, as shown in Figure 9a. The blade dynamic system shows a LCO response at a superposed motion with different frequencies and amplitudes in the phase plane, associated with different nonlinear behaviors of the aeroelastic system. This appearance of chaos is related to the existence of multiple frequency components in the blade vibration under a high flow condition. In this state the frequency of non-synchronous begins to emerge with a low amplitude.

Then the magnitude of blade oscillation increases as well as the LCO amplitude with the decrease of flow rate and the amplitude of the characteristic frequency rises rapidly to the dominant status. The topological structure of the system response is found to change gradually. With the entry of non-synchronous vibration, the limit cycle oscillation occurs in the system and the aeroelastic response of the blade presents a quasi-periodic state indicated in Figure 9b. In this second stage, the oscillation amplitude of the system maintains the LCO response with a high value. The behavior of quasi periodic oscillation is corresponding to the dominant frequency of blade vibration. The performance of the dynamic system is consistent with the phenomena of non-synchronous vibration.

In the third stage, the oscillation amplitude in phase plane decreases continuously with the reduction of blade vibration. The response of the system gradually develops from the single period motion to a multi-period motion. With the exit of the NSV, the blade oscillation expresses a stable state with a low vibration amplitude. It can be observed that the system exhibits a convergent aeroelastic response.

Under the combined action of aerodynamic force and damping, the blade vibration exhibits the LCO response. The response of nonlinear dynamic system can be divided into three kinds of behaviors, as chaos, LCO and the multi periodic motion. Correspondingly, the whole vibration process of blade in NSV yet can be regarded as three stages, as origination, development and disappearance.

## 4. Study on the Mechanism and Parametric Description for Non-Synchronous Vibration

From the analysis above, it can be seen that the main features of non-synchronous blade vibration phenomenon in a high-speed compressor can be captured through time-domain fluid structure coupling simulations. Next, a deep investigation on the mechanism of formation is carried out in an attempt to describe the parameters for NSV.

### 4.1. Evolution of Flow Structure in Field

In order to observe the flow phenomena in different states, instantaneous distributions of local flow are sketched based on the computation results. As indicated in [23], the flow entered into the blade passages along the flow path smoothly to the exit when the rotor is operating far away from the fault state. However, in the streamline diagram on the 90% span of the blade, an inverse flow appears in the passage when approaching to the NSV point. Figure 10a shows that a backflow begins to occupy the main part of region in the blade tip. The main flow could not continue to flow smoothly but with a certain amount of resistance in passage 1. It is worth noting that the appearance of the backflow is alternating in passages.

With the flow rate decreasing further, the blockage near the blade suction side accumulates sharply at the tip trailing edge in passage 1 and a countercurrent flow occurs as a part of flow traverses the rotor outlet. This is clearly identified by a red line drawn in Figure 10b. With the movement of the reversal flow, the blockage at tip region is pushed forward to the leading edge of blade. Interestingly, there is a significant difference in the adjacent channel. A swirl of vortex is formed at suction side of blade trailing edge, which gives rise to an obvious increase in entropy. This appearance of swirling flow is consistent with the occurrence of non-synchronous vibration. And it is considered to be the inducing factor for NSV.

Finally, this kind of swirling vortex dissolves away in the flow instead of the separation flow which is pushed out of the blade passage. In the circumferential direction, the interface of the main flow and the separation vortex in the passage is in coincidence with the leading edge of the adjacent blades at inlet. This flow phenomenon is displayed in Figure 10c. In the meantime, the blade vibration is out of the non-synchronous state. It can be predicted that the rotating stall will occur in the next.

### 4.2. Inducement of Non-Synchronous Vibration

It can be concluded that the unsteadiness of the complex flow plays an important role on the separation backflow. This swirling backflow is the direct cause that puts the blade in the NSV motion. The blockage near the casing is accumulated due to the mixture of separation vortexes and tip leakage flows. The interaction of the main flow and the local separation flow is responsible for the periodical swirling vortex in the passages, which is revealed in Figure 10b. For the purpose of recognizing the structure of the flow, the streamlines in the blade tip at state C are illustrated from the perspective of three-dimensions in Figure 11 where the helical vortex structure is observed at the leading edge.

The observed vortex swirls clearly at the blade suction side in the tip region. Compared with the chaos of the general separated flow, the swirling vortices appear to be “orderly” in this study. It can be seen that, affected by adverse pressure gradient, the leakage flow is pushed upstream. Then this leakage flow enters into the adjacent channel under the action of the main flow, but ablocked by the flow separation at the outlet of the adjacent channel, it returns back to the inlet once more. During the process, the separation vortices on the suction blade present to be the spiral vortex, and flow to the leading edge of the adjacent blade. Mixed up with the main flow of the adjacent passage, the flow continues to form another spiral vortex in the circumference.

In order to discover the relationship between the spiral vortex and blade vibration, the static pressure contour near the blade tip is analyzed. The field in the rotor tip is depicted in time-sequences to show the periodic evolution of the flow structure. The region of low pressure corresponds to the spiral vortex. It can be seen from the Figure 12 that the spiral vortex located in the leading tip rotates along the direction of rotation. According to the computations it is divided into 12 steps in a cycle of non-synchronous vibration. At the beginning of evolution, there is a location of swirling vortex at leading edge of the blade. As time goes on, this swirling vortex happens to shed off, and transfers to the adjacent blade shown in time of 3/12 *T_nsv_*. Finally, the rotating blade catches up the vortex of the adjacent blade tip in time of 6/12 *T_nsv_*. In the rotor rotation, the blade continuously pursues the vortex in front.

The results show that the disturbances of spiral vortex near the blade tip propagates in the annulus, indicating a pattern as modal oscillation. Composed of the separation vortex shedding and tip leakage flow, the mixed vortices lead to a large fluctuation of the local pressure. As the rotor rotates, the blades are constantly sweeping the vortex, experiencing the oscillation of disturbances. This oscillating phenomenon results in another aerodynamic fluctuation acted on the blade. When these separation vortexes become powerful enough to alter the blade loading, the blade may vibrate with a large amplitude. If the natural mode of the blade coincides with the frequency of sweeping vortex, the blade vibration in non-synchronous state will be stimulated.

### 4.3. Parametric Description for Non-Synchronous Vibration

From the analysis above, we can see that there are obvious discrepancies in the process of entry and exit of NSV. The disturbances of spiral vortex in the tip are considered to be the inducing factor on the oscillation of blade loading. Therefore, the strength of swirling separation flow is of great significance for leading the non-synchronous vibration of blade. In this section, the parameterization of the inducing factor is preliminarily explored for the NSV.

With the decrease of the flow rate, the interface of the reversal flow with the main stream moves forward, which indicates an increasing intensity of the separated leakage flow shown in Figure 10. With the continuous development of vortex, the spiral vortices appear in the circumference. Because the mixing of leakage flow and separation flow in the spiral vortex is a main cause of NSV, it is necessary to quantitatively analyze and judge the inducing condition of the clearance flow in the process of blade vibration. Figure 13 shows the flux ratio of leakage flow to the total mass flow under different working conditions. There is a rapid rise on the point of inflection at state B. The ascendant ratio means the strength of swirling separation leakage flow is increased because of adverse pressure gradient. And the flow field structure also changes at this moment.

The axial momentum is used to quantitatively describe the intensity of the reverse leakage flow. This is the equation of axial momentum for the tip flow:(2)$$M_{Z}=\int_{0}^{z}\int_{0}^{\sigma }\rho \left(v\cdot n\right)wdrdz$$
where *z* is the axial coordinate, *σ* is the tip clearance height, *ρ* is the flow density, *w* is the axial velocity, **v** is the velocity vector and **n** is the normal vector. As indicated in Figure 12, the swirling flows in adjacent blade passages exist differences. The oscillation of blade loading occurs with blade rotating pass these disturbances. Therefore, the discrepancy of axial momentum of the reverse leakage flow in adjacent channels plays a crucial role in the blade vibration.

The variation of axial momentum ratio is revealed by Figure 14 for the reverse leakage flow in two adjacent channels under different working conditions. It can be seen that there is an obvious jump in the curvature above a certain value. For the compressor studied in this paper, the blade vibration happens to be in non-synchronous state in the range of 1.7–2.3. When located near the ratio of 1, the rotor has been out of the state of NSV. In this moment, the discrepancy of flow in adjacent passages is very small. The amplitude of oscillation is also reduced so a large range of pressure fluctuation is the premise of blade vibration. When the ratio is close to the location of 2, this precondition is satisfied. Therefore, the axial momentum of the reverse leakage flow in adjacent channels can be chosen as a parametric description for NSV evaluation.

## 5. Discussions

It is concluded that the spiral vortex near the tip is considered as a main inducing factor for the presented NSV. This kind of flow instability is composed of the separation vortex shedding and tip leakage flow. The periodical swirling vortex located in the tip is observed to exist in every passage with different strengths and rotates along the direction of rotation. However, it is necessary to make a distinction between the swirling vortex and rotating stalls. First, there is not a sharp decrease in the performance. Secondly, this kind of spiral vortex disappears in the passage with the decrease of flow rate. Interestingly, the blade vibration exits from the state of NSV correspondingly. This is not common in previous literatures. Investigation on the withdrawal mechanism of NSV deserves further effort.

It can be observed that the oscillation of blade loading is related to the instability analyzed above. The differences of flow field in adjacent passages result in the pressure fluctuation when blades are passing these disturbances. This indicates that the blade vibration in the present investigation is a typical flow-induced vibration coupled to the tip flow instability. Besides the frequency of excitation, incentive intensity is another important factor. The ratio of axial momentum of the reverse leakage flow in adjacent channels is analyzed quantitatively during the initiation and development of NSV. And The parameterization for the NSV is worthy of further analysis.

## 6. Conclusions

In order to explore mechanism during the development of the non-synchronous vibration, a time domain numerical approach was adopted using the fluid–structure iterative coupling method. The main features of non-synchronous blade vibration are captured with a description on the nonlinear dynamic system of blade. The origin and development of the entropy flow for non-synchronous vibration are discussed in detail. The main conclusions can be summarized as follows:

(1) Under the current methodology the vibration of rotor blades in the first stage is found to be in the first bending mode with a non-integral order (4.6) of the rotating speed. This characteristic frequency appears to be in dominant status with a developing process. The average IBPA is maintained a close value of 151.5° as the rotor enters into NSV state. The angle decreases rapidly when the blades bow out of the state. Referring to the constant IBPA corresponding to the modal vibration, the appearances of frequency-locking and phase-locking can be identified for the NSV with a non-integral order.

(2) Due to the fluid and structure nonlinearities, the blade vibration exhibits the LCO response. The response of a non-linear dynamic system can be divided into three kinds of behaviors, namely chaos, LCO and the multiperiodic motion. Correspondingly, the whole vibration process of blade in NSV can also be regarded as consisting of three stages, that is origination, development and disappearance.

(3) The blockage near the casing is accumulated due to the mixture of separation vortex and tip leakage flow. The interaction of main flow and the local separation flow is responsible for the periodical swirling vortex in the passages. The results show that the disturbances of spiral vortex near the blade tip propagate in the annulus, indicating a pattern of modal oscillation. This oscillating phenomenon results in another aerodynamic fluctuation acting on the blade. A flow-induced vibration is initiated by the spiral vortex in the tip and it is regarded as a main inducing factor for the presented NSV.

(4) The ratio of axial momentum of the reverse leakage flow in the adjacent passage is chosen as a parametric description for NSV evaluation. The NSV is observed to happen in the range of 1.7–2.3. This means that the swirling vortices in adjacent passages exist an obvious diversity of strengths, which can result in a pressure fluctuation when the blades are passing these disturbances.

## Figures and Tables

**Figure 1 entropy-23-00383-f001:**
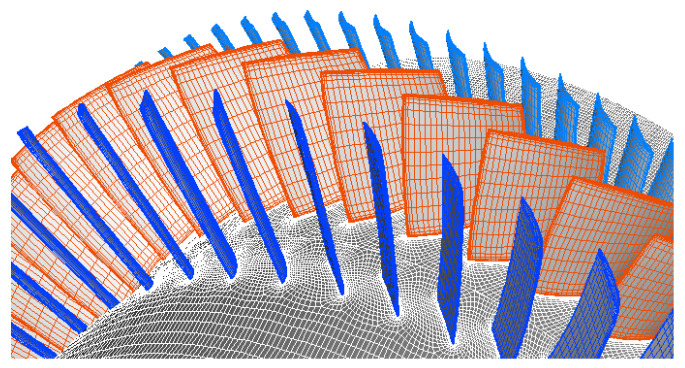
Numerical model of 1.5 stage high speed compressor.

**Figure 2 entropy-23-00383-f002:**
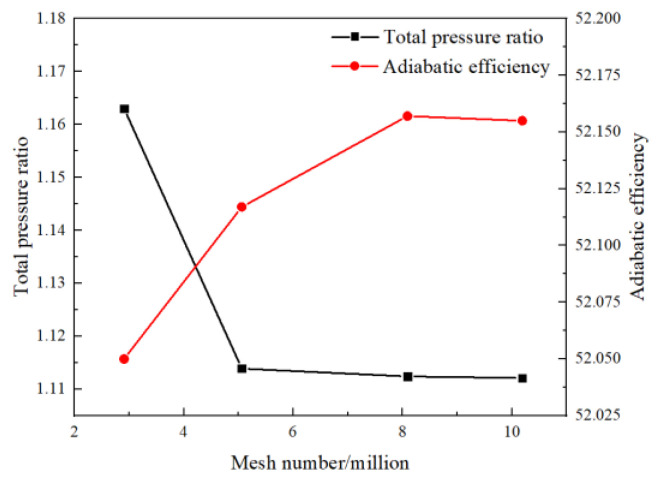
Performances with different grid density.

**Figure 3 entropy-23-00383-f003:**
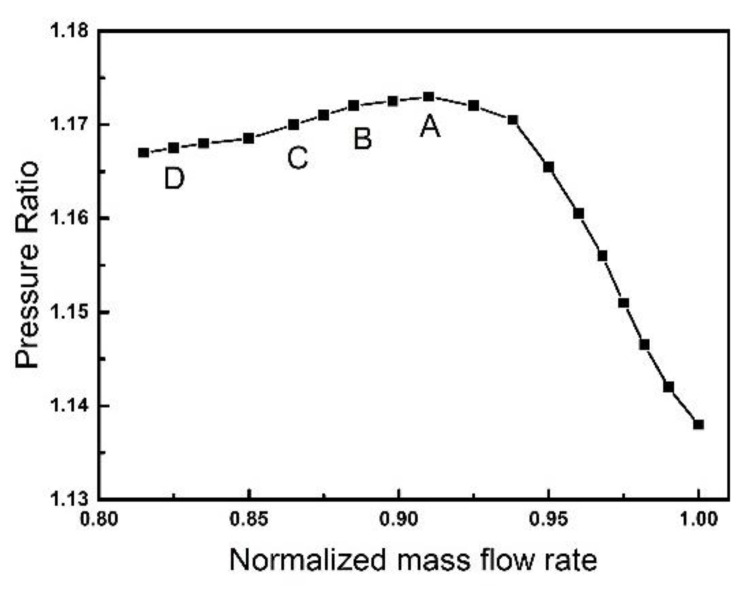
Operating states in computation.

**Figure 4 entropy-23-00383-f004:**
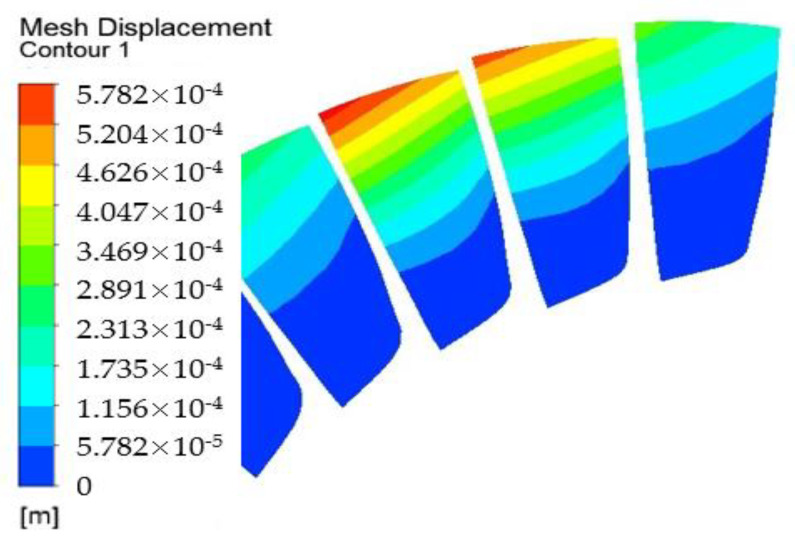
Deformations of blade vibration at state C.

**Figure 5 entropy-23-00383-f005:**
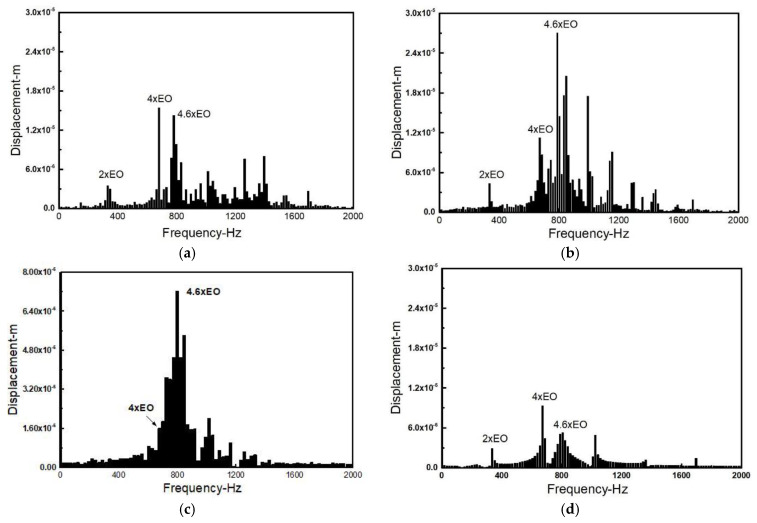
Spectrum diagrams of blade vibration: (**a**) State A; (**b**) State B; (**c**) State C; (**d**) State D.

**Figure 6 entropy-23-00383-f006:**
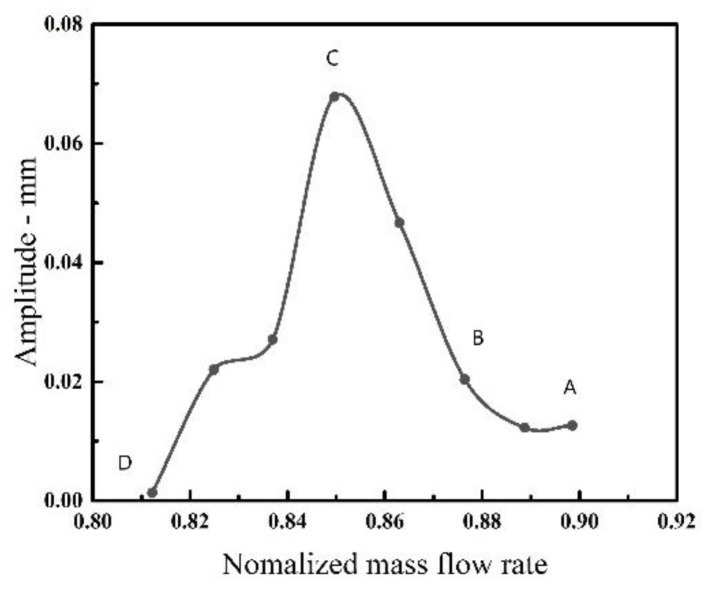
Amplitude variation of blade vibration.

**Figure 7 entropy-23-00383-f007:**
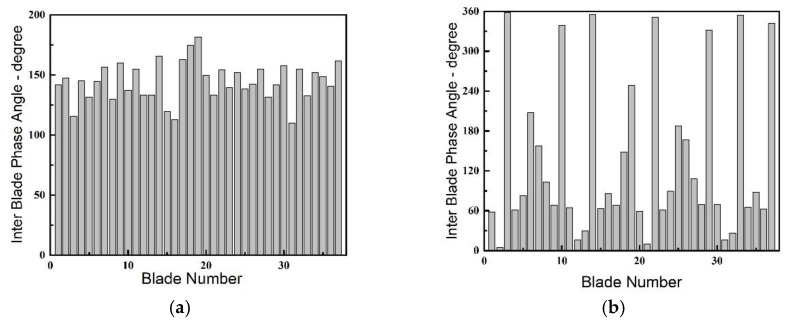
Inter blade phase angles in vibration: (**a**) State B; (**b**) State D.

**Figure 8 entropy-23-00383-f008:**
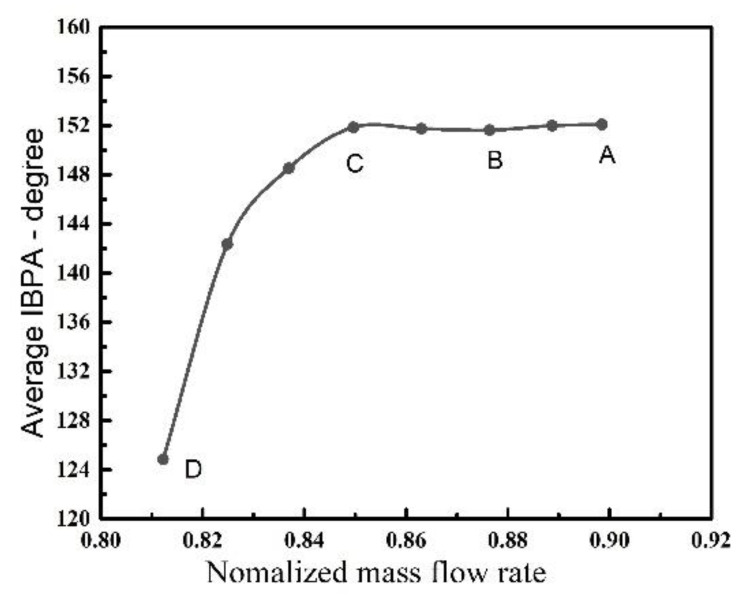
IBPA variation during vibration.

**Figure 9 entropy-23-00383-f009:**
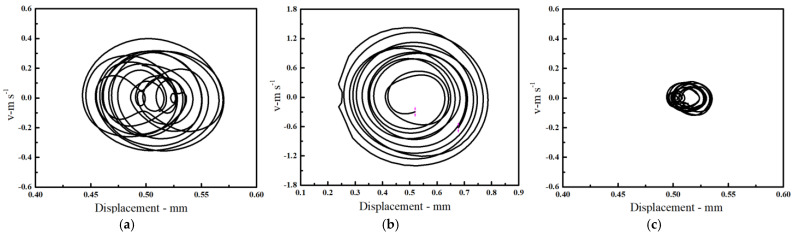
Phase diagrams of blade vibration: (**a**) State A; (**b**) State C; (**c**) State D.

**Figure 10 entropy-23-00383-f010:**
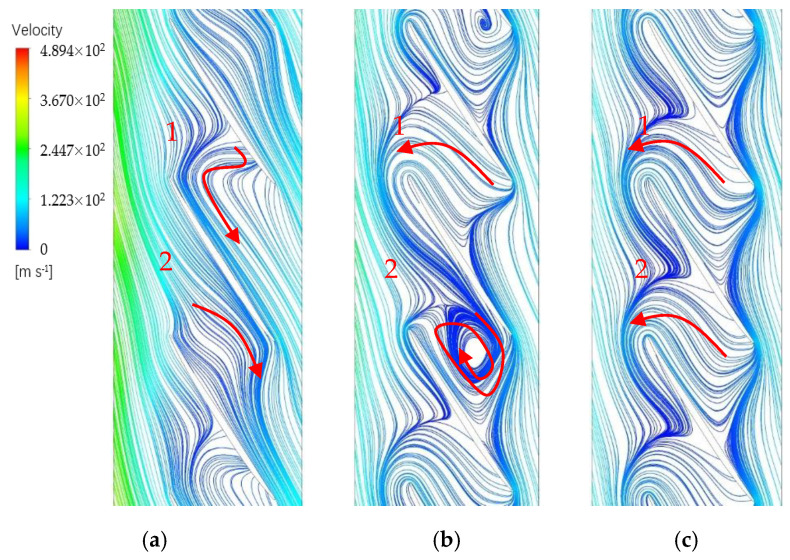
Streamline diagrams at the 90% span of blade: (**a**) State A; (**b**) State C; (**c**) State D.

**Figure 11 entropy-23-00383-f011:**
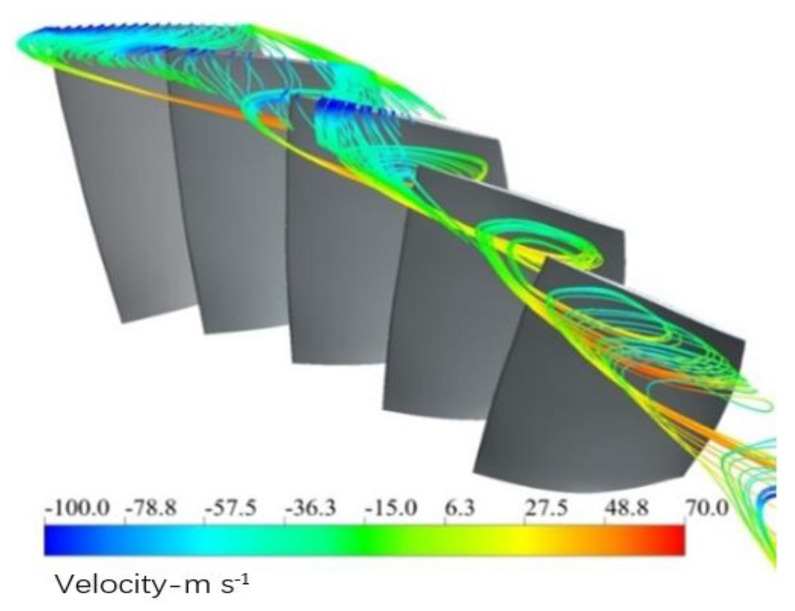
Spiral vortex at the leading edge in the blade tip.

**Figure 12 entropy-23-00383-f012:**
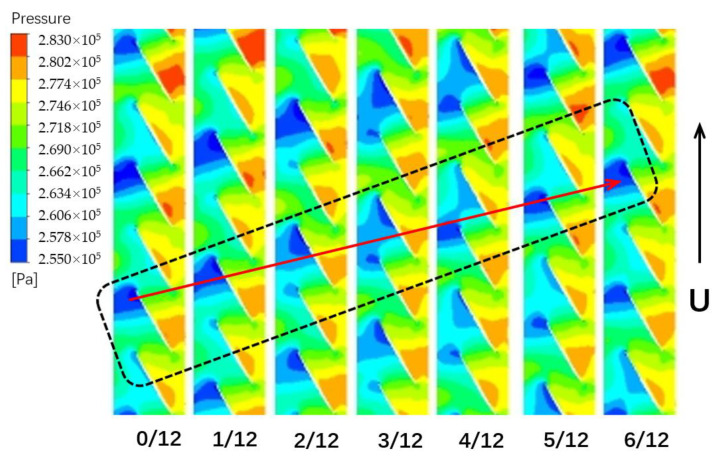
Pressure contours of the spiral vortex at the 90% span of blade.

**Figure 13 entropy-23-00383-f013:**
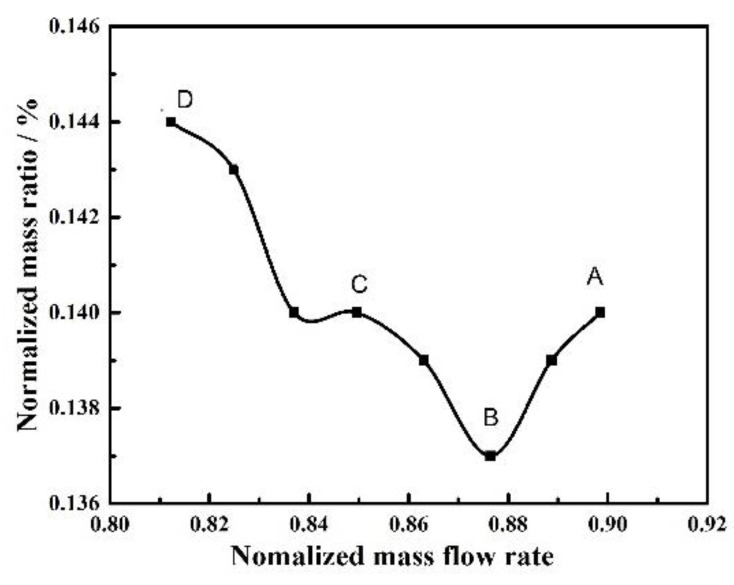
Flux ratio of leakage flow to the total massflow.

**Figure 14 entropy-23-00383-f014:**
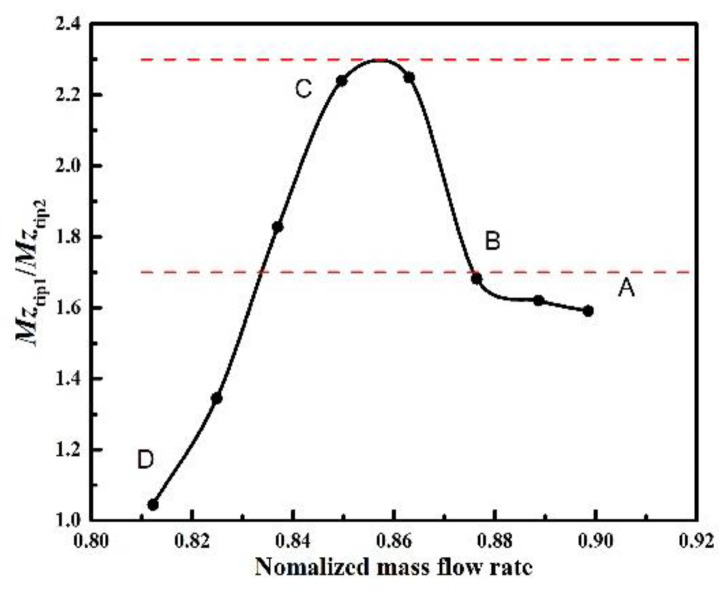
Variation of axial momentum ratio in the blade tip.

**Table 1 entropy-23-00383-t001:** Distribution of grids.

Zone	Domain	Mesh Size
IGV	42	258.5 × 10^4^
Rotor	38	407.7 × 10^4^
Stator	82	142.9 × 10^4^
Total		809.1 × 10^4^

## Data Availability

Data sharing not applicable.

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
