# Peer review of "Investigation on the Mechanism and Parametric Description of Non-Synchronous Blade Vibration"

_entropy, 2021, doi:10.3390/e23040383_

Round 1
Reviewer 1 Report
This paper presents a numerical study of non-synchronous blade vibration in a 1.5 stage high speed compressor. On the studied compressor, the inception, development and disappearance of NSV are identified, and physical phenomena are proposed to explain the observations. The spiral vortex is recognized as the main phenomenon inducing NSV on this configuration. This paper is a valuable contribution in the research field of NSV. However, additional validation against experimental data would be beneficial to the paper.
Major remarks
The compressor is not presented in the paper, only a reference is cited. It would however be relevant to add more details on the compressor, especially on the first 1.5 stage. For instance, is it subsonic or transonic?
Due to rather large values of y+, a wall law must be used. Could the authors checked that this is indeed the case, and provide some more details on this wall law?
In addition to Fig.2, it would be relevant to compare 1D data, such as radial profiles at the outlet of the compressor, to assess the mesh convergence study.
The only validation against experimental data can be found in 3.1, regarding a single value of bending mode. It would be very beneficial to add other comparisons with experimental data, for the reader to have confidence in the numerical results presented.
Fig.4: at which operating point does the Fig. correspond?
Fig. 11: at which operating point does the Fig. correspond?
Could the authors provide figures of the tip leakage vortex, from points A to D? The spiral vortex identified by the authors seems to appear when the tip leakage vortex goes out from the blade passage, i.e. at point A or close to it. It would be interesting to see if this is the case.
Minor remarks
English language is not of the quality level expected for a journal paper, please review it extensively.
p.2 : “The aim of current paper is to explore the origin and development of the entropy flow for blade vibration, discuss the parametric description for NSV.” This sentence is not clear, please reformulate.
Could the authors precise how is computed the IBPA?
Author Response
Thank you very much for the suggestions for our draft. We have replied point by point to your comments, along with a clear indication of red color at the location of the revised paper.
The comments and replies can be summarized as follows:
- Q: The compressor is not presented in the paper, only a reference is cited. It would however be relevant to add more details on the compressor, especially on the first 1.5 stage. For instance, is it subsonic or transonic?
- A: The computational domain of the 1.5 stage compressor is shown in figure 1. The investigated object is a transonic compressor, which has been supplemented in section 2. To avoid repetitive expression, more detailed descriptions of the compressor are cited in our previous work published in Entropy, as shown in reference [23].
- Q: Due to rather large values of y+, a wall law must be used. Could the authors checked that this is indeed the case, and provide some more details on this wall law?
- A: The values of y+ has been checked, which is smaller than 20 near the wall boundary. Thus, the scalable wall function in CFX software is applied to resolve the flow within boundary layer, and it has been supplemented in the paper. Because the wall function is commonly used, the details are not shown in the paper, which can be found in CFX software.
- Q: In addition to Fig.2, it would be relevant to compare 1D data, such as radial profiles at the outlet of the compressor, to assess the mesh convergence study.
- A: In the work, total pressure and adiabatic efficiency are selected to validate the mesh independence. According to figure 2, the mesh of 8.09 million has been proved to be qualified. In addition, the radial distribution of pressure at the outlet of rotor is more complicated. So it is not used in this paper.
- Q: The only validation against experimental data can be found in 3.1, regarding a single value of bending mode. It would be very beneficial to add other comparisons with experimental data, for the reader to have confidence in the numerical results presented.
- A: The validation of simulation has been detailed checked in previous paper (Investigation on the Flow Field Entropy Structure of Non-Synchronous Blade Vibration in an Axial Turbocompressor. Entropy 2020,22,1372.), including vibration frequency, vortex frequency and IBPA. So in present paper, this part is weakly emphasized. But at the beginning of part 3, the sentence of “As the reliability and accuracy of the numerical method are verified in the reference [23] earlier, the comparison with the experiment demonstrates that it is sufficient to predict the NSV phenomenon by the current methodology.” is expressed.
- Q:4: at which operating point does the Fig. correspond?
- A: Figure 4 displays the blade vibration deformation at state C, which has been supplemented in the revised paper.
- Q: 11: at which operating point does the Fig. correspond?
- A: Figure 11 displays the streamlines in the blade tip at state C, which has been supplemented in the revised paper.
- Q: Could the authors provide figures of the tip leakage vortex, from points A to D? The spiral vortex identified by the authors seems to appear when the tip leakage vortex goes out from the blade passage, i.e. at point A or close to it. It would be interesting to see if this is the case.
- A: Indeed, this kind of flow instability is composed of the separation vortex shedding and tip leakage flow. But it hard to tell whether the flow instability appears when the leakage vortex goes out from the passage. Because of the limitation of k-εturbulence model, the leakage flow is clearly distinguished with separation vortex until the state C.
- Q: English language is not of the quality level expected for a journal paper, please review it extensively.
- A: The typographical and grammatical errors had been revised most in the revised paper. We have made the correction accordingly.
- Q:2 : “The aim of current paper is to explore the origin and development of the entropy flow for blade vibration, discuss the parametric description for NSV.” This sentence is not clear, please reformulate.
- A: The work focuses on finding the inducement of blade vibration and attempting to describe NSV parametrically. Within the author's understanding, a high entropy in filed means the flow loss, which is caused by the separation vortices. So an entropy structure here refers to the structure of the flow loss, especially the vortices leading to the blade vibration. The sentence has been revised as “The aim of current paper is to explore the origin and development of the entropy flow which causes blade vibration, and discuss the parametric description for NSV.”
- Q: Could the authors precise how is computed the IBPA?
- A: By applying Fast Fourier transform to the blade vibration displacement data, the phase of the blade vibration can be obtained, and the IBPA is the phase difference of the adjacent blades.
Thanks again for your advice.

Reviewer 2 Report
This paper presents a numerical study with fluid-structure interaction and investigates the vibrations of blades under selected scenarios.
What is missing is a detailed description of modeling assumptions (for example to add in Section 2) and support all the following results that are discussed in the manuscript (boundaries, loads etc).
The discussion of results needs also additional input to properly introduce al the graphical items and make clear what do they represent and how they are derived from simulations.
Author Response
Thank you very much for the suggestions for our draft. We have replied point by point to your comments, along with a clear indication of red color at the location of the revised paper.
The comments and replies can be summarized as follows:
- Q:What is missing is a detailed description of modeling assumptions (for example to add in Section 2) and support all the following results that are discussed in the manuscript (boundaries, loads etc).
- A: The computational domain of the 1.5 stage compressor is shown in figure 1. The investigated object is a transonic compressor, which has been supplemented in section 2. To avoid repetitive expression, more detailed descriptions of the compressor are cited in our previous work published in Entropy, as shown in reference [23].
The validation of simulation has been detailed checked in previous paper (Investigation on the Flow Field Entropy Structure of Non-Synchronous Blade Vibration in an Axial Turbocompressor. Entropy 2020,22,1372.), including vibration frequency, vortex frequency and IBPA. So in present paper, this part is weakly emphasized. And at the beginning of part 3, the sentence of “As the reliability and accuracy of the numerical method are verified in the reference [23] earlier, the comparison with the experiment demonstrates that it is sufficient to predict the NSV phenomenon by the current methodology.” is expressed.
- Q:The discussion of results needs also additional input to properly introduce all the graphical items and make clear what do they represent and how they are derived from simulations.
- A: For the sake of exploring the origin and development of NSV, the analysis on the vibration characteristics of compressor blade is performed under different working states (Fig.3). The NSV doesn’t appear in a suddenness, but with a developing process (Fig.5). The amplitude reaches the peak at state C (Fig.6).
The average IBPA is maintained a close value (Fig.7a), as a wave superimposed on the rotor. The blade vibration exhibits the LCO response, which can be divided into three kinds of behaviors (Fig.9).
The blockage near the casing is accumulated due to the mixture of separation vortex and tip leakage flow (Fig.10). The disturbances of spiral vortex near the blade tip propagate in the annulus, indicating a pattern as modal oscillation (Fig.11). And the NSV is initiated by the spiral vortex in the tip (Fig.12).
In order to quantitatively analyze and judge the inducing condition in the process of blade vibration, the flux ratio of leakage flow to the total massflow under different working conditions is shown (Fig.13). It is found that there is an obvious jump in the curvature above a certain value (Fig.14), corresponding to the non-synchronous state.
The typographical and grammatical errors also had been revised most in the revised paper. We have made the correction accordingly.
Thanks again for your advice.

Round 2
Reviewer 1 Report
Following the answers of the authors, the paper is now ready for publication.